# Predictors of loss to follow up from antiretroviral therapy among adolescents with HIV/AIDS in Tanzania

**Esther-Dorice Tesha** [1,2]* , **Rogath Kishimba**[2‡], **Prosper Njau**[3‡], **Baraka Revocutus**[3‡], **Elia Mmbaga**[1,4]

**1** Department of Epidemiology and Biostatistics at Muhimbili University of Health and Allied Sciences, Dar es Salaam, Tanzania, **2** Tanzania Field of Epidemiology and Laboratory Training Program, Dar es Salaam, Tanzania, **3** National AIDS Control Program, Ministry of Health, Community Development, Gender, Elderly, and Children, Dodoma, Tanzania, **4** Department of Community Medicine and Global Health, University of Oslo, Oslo, Norway

☯ These authors contributed equally to this work.
‡ These authors also contributed equally to this work.
* drespick@yahoo.com

**Data Availability Statement:** This study used the data set from CTC3 database from NACP combining all patients' records attending CTC clinic. Permission to access the data was granted

## Abstract

Access to Antiretroviral Therapy (ART) is threatened by the increased rate of loss to follow-up (LTFU) among adolescents on ART care. We investigated the rate of LTFU from HIV care and associated predictors among adolescents living with HIV/AIDS in Tanzania. A retrospective cohort analysis of adolescents on ART from January 2014 to December 2016 was performed. Kaplan-Meier method was used to determine failure probabilities and the Cox proportion hazard regression model was used to determine predictors of loss to follow up. A total of 25,484 adolescents were on ART between 2014 and 2016, of whom 78.4% were female and 42% of adolescents were lost to follow-up. Predictors associated with LTFU included; adolescents aged 15–19 years (adjusted hazard ratio (aHR): 1.57; 95% Confidence Interval (CI); 1.47–1.69), having HIV/TB co-infection (aHR: 1.58; 95% CI, 1.32–1.89), attending care at dispensaries (aHR: 1.12; 95% CI, 1.07–1.18) or health center (aHR: 1.10; 95% CI, 1.04–1.15), and being malnourished (aHR: 2.27; 95% CI,1.56–3.23). Moreover, residing in the Lake Zone and having advanced HIV disease were associated with LTFU. These findings highlight the high rate of LTFU and the need for intervention targeting older adolescents with advanced diseases and strengthening primary public facilities to achieve the 2030 goal of ending HIV as a public health threat.

## Introduction

In 2018, an estimate of 1.6 million adolescents aged 10 to 19 were living with HIV worldwide [1]. World Health Organization (WHO) reported HIV counseling and testing, linkage to care and treatment to be a challenge [2]. Of all patients enrolled in HIV care, only 23% were retained [3]. A review done in Sub-Saharan African countries showed the rate of loss to follow up (LTFU) among adolescents ranged between 15% and 54% [4]. In 2015, Tanzania National

by Ministry of Health Tanzania under the National AIDS Control Program (third party). The author did not have the right to share the data publicly. In order to gain access to data kindly contact: National AIDS Control Programme-NACP P O Box 784 Dodoma, Tanzania Kilimani Area, NACP/ NTLP Building. Tel: +255 (0) 262060148 E-mail: nacp@afya.go.tz.

**Funding:** The author received no specific funding for this work.

**Competing interests:** The authors have declared no competing interest or personal relationship that could have appeared to influence the reported work.

AIDS Control Program (NACP) reported the rate of LTFU among 75, 596 patients aged 15–24 years to be 23.5% [5]. Similarly, a study in Nigeria reported the probabilities of LTFU among adolescents living with HIV (ADLHIV) were 3.6%,6.9,% and 35.9% at 6,12, and 25 months respectively [6]. Predictors associated with increased risk of LTFU among ADLHIV reported in various studies included; ADLHIV aged 15–19 years, female adolescents, those diagnosed with HIV/TB co-infection, those with malnutrition, adolescents who attended clinics at primary facilities and having advanced WHO clinical stage [7–12]. Also increased risk of LTFU was observed among adolescents who had prior exposure to ART [13] and those who attended clinics at public health facilities [14].

In Tanzania, various studies and routine assessments of HIV services have been done among adults but limited data exist among adolescents group [15]. Routine data indicates a high rate of loss to follow-up among adolescents globally and in Tanzania. To achieve the 2030 goal of ending the HIV epidemic as a public health threat, identification of predictors of LTFU on ART is urgently needed to inform effective strategies of retention in care among adolescents living with HIV/AIDS.

## Methods

### Study design and population

We performed a retrospective cohort study of routinely collected data obtained from the Care and Treatment Clinics (CTC3) macro database at the National AIDS Control Program (NACP). This study included adolescents who were enrolled and initiated ART at age 10–19 with either vertical or horizontal HIV/AIDS infection from January 2014 to December 2016. Adolescents whose records had missing details of enrollment date and last appointment date in the CTC3 macro database were excluded from the study. Adolescents who were transferred out and died were censored at the time of their last visit.

### Study setting

The data involved all ADLHIV enrolled in ART care from 2014 to 2016 in all 26 regions of Tanzania mainland available in CTC3 database. Tanzania mainland is the largest country in East Africa covering 947,300 square kilometers with an estimate of 55 million population. In 2020, population of adolescents aged 10–19 years in Tanzania mainland was projected to be 13,206,921 based on the 2012 national census [16] with the HIV prevalence of 5.8% [17]. In Tanzania, adolescents can access ART care in health facilities which are categorized into three levels. These levels include; primary health facilities also known as dispensaries, secondary health facilities (health centers), and tertiary health facilities (hospitals). The Tanzania HIV guideline requires adolescents to attend HIV clinics at least once a month for the first three months of the ART treatment and thereafter once every three months depending on their adherence status [18]. To reduce the burden of TB in HIV patients all ART clinics are integrated with TB/HIV services [18]. TB/HIV Co-infection refers to HIV patients who were also diagnosed with TB infection during routine TB screening. In Tanzania, all people living with HIV (PLHIV) are screened for TB on every clinic visit, and those not infected are provided with Isonized Preventive Therapy (IPT) to prevent them from developing active TB.

### Data source

Data were extracted from CTC3 database at NACP combining all patients' records attending CTC.

Standard patient management form known as CTC2 cards are used to capture patients' records in the facilities. The CTC2 cards have information on; patient's socio-demographic characteristics, WHO clinical stage, weight, CD4 counts, ART regimen, ART initiation date, test results (TB screening), nutrition status, height, ART adherence status, mortality records, and haemoglobin level. In each CTC clinic, patients' records are directly exported to the CTC3 database through a computer. All patients are recorded by their unique identification number at the time of enrolment.

## Study variables

The main outcomes in this study were loss to follow-up from HIV care (ART) and retention on ART care. The loss to follow-up was defined as any of the adolescents who failed to attend at least one clinic visit within 90 days of their scheduled appointment. Retention to care was defined as the state of an adolescent being alive, actively attending scheduled appointments at the clinics, and receiving ART care at the end of follow-up period. Independent variables of the study included; sex which was classified as male and female, and age was grouped as 10–14 years and 15–19 years. Marital status was grouped as never married/single and cohabiting/married, types of health facilities were categorized as dispensary, health centers, and hospitals. Also, facility ownership was classified as public and private, WHO clinical stages were grouped into stage I, II, III, and IV. Geographical zones were grouped as coastal, central, lake, northern, southern highland and western zones.

Nutrition status was categorized into two groups according to BMI scales. BMI was obtained from adolescent's weight in kilograms divided by the square of height in meters which were assessed at every clinic visits. Participants with BMI of $<18.5kg/m^2$ (underweight) or $25.0–29.9kg/m^2$ (obesity/overweight) were classified as "malnutrition", while those with BMI between $18.5–24.9kg/m^2$ (normal weight) were classified as "no malnutrition".

ART regimens were categorized as first-line and second-line based on Tanzania HIV guideline [18]. First-line regimens included; Tenofovir (TDF) 300 mg / Lamivudine (3TC) 300 mg / Efavirenz (EFV) 600mg. Alternative first line regimen used were; Tenofovir (TDF) + Emtricitabine (FTC) + Dolutegravir (DTG), Abacavir (ABC) + Lamivudine (3TC) + Efavirenz (EFV) and Zidovudine (AZT)+Lamivudine(3TC) +Nevirapine (NVP). Also, the second-line regimens included; Zidovudine (AZT), Tenofovir (TDF), Abacavir (ABC), Lamivudine (3TC), Emtricitabine (FTC), Atazanavir boosted by Ritonavir (ATV/r), Lopinavir boosted by Ritonavir (LPV/r) and Dolutegravir (DTG). Other independent variables included; prior exposure to ART during enrolment (yes or no) and HIV/TB Co-infection (yes or no).

## Statistical analysis

Stata 15 IC (StataCorp, College Station, TX, USA) was used for analysis. Patients' characteristics were summarized using proportion for categorical variables, mean and standard deviation for continuous variables. Overall and covariate specific rate of loss to follow up per 1000 person-months (pm) was determined. Kaplan-Meier curves were used to determine failure probabilities among adolescents who were enrolled on ART. The log-rank test was used to test for statistical significance of the difference in the failure probability curves. Predictors of loss to follow up on ART were determined using Cox proportion hazard regression model. Variables with a p-value $\leq 0.2$ in the bivariate analysis and other potential confounders were included in multivariate analysis. Crude and adjusted hazard ratios with their respective 95% confidence interval were reported.

## Ethical consideration

The study and analyses of the CTC3 patients' records were covered by ethical approval from Muhimbili University of Health and Allied Sciences (MUHAS) with Ref. No. DA.287/298/ 01A. Since this study used secondary data obtained from the CTC3 database available at the Tanzania National AIDS Control programs, the institutional review board (IRB) waived the need for consent from parents and guardians of the adolescents. The informed consent was not required because patients' records were anonymized before access and adolescent's records were identified by their unique registration numbers. However, permission to access data was obtained from the National AIDS Control Program.

## Results

A total of 25,880 records were obtained from the CTC-3 database. About 396(1.5%) records with no ART enrollment and last appointment date were excluded from this study. Of these, 25,484 (98.5%) records were eligible for this study based on inclusion criteria (Fig 1). About 177(0.7%) records were deceased and 74(1.7%) records were transferred out on their last appointment date. A total of 14,498(56.8%) records were retained duringfollow-uow up time of the study.

### Baseline characteristics of the participants

Three-quarter (78.36%) of the adolescents on ART were female. Above seventy percent (72.09%) of the study cohort were aged between 15–19 years with a median (Interquartile range, IQR) age of 17 (14–18) years. At enrollment over ninety percent of adolescent had no HIV/TB Co-infection and more than one-third (45.96%) were in WHO stage I. Adolescents who were on the first line regimen accounted for 98.84% of the study population. Above 90% of adolescents on ART were termed to have normal nutrition status (Table 1).

### Rate of loss to follow up among adolescents on ART

The follow-up time of 8,100 person-months at risk (pmr) was accumulated from 25,484 adolescents who were on ART during the study period. A total of 10,735 (42.20%) were LTFU

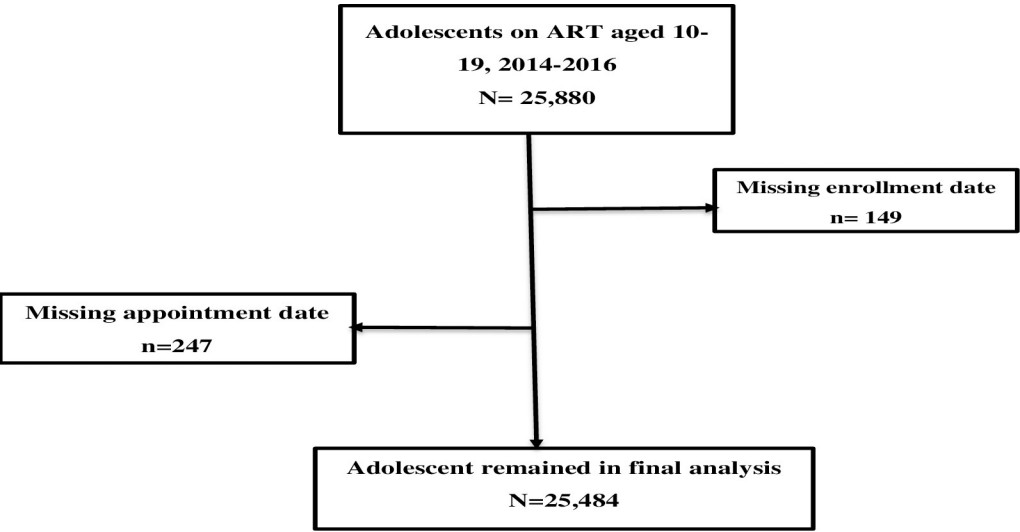

**Fig 1. Flow chart for the selection procedure of the study subjects.**

**Table 1. Characteristics of adolescents on ART (N = 25,484).**

| Characteristics | Number | Percentage |
|---|---|---|
| **Sex** | | |
| Male | 5,515 | 21.64 |
| Female | 19,969 | 78.36 |
| **Age groups (years)** | | |
| 10–14 | 7,112 | 27.91 |
| 15–19 | 18,372 | 72.09 |
| **Median (IQR) age** | 17(14–18) | |
| **Marital status** | | |
| Never married/Single | 12,963 | 50.87 |
| Cohabiting/Married | 7,701 | 30.22 |
| Missing | 4,820 | 18.91 |
| **Types of Health Facility** | | |
| Dispensary | 9,114 | 35.76 |
| Health Centre | 8,573 | 33.64 |
| Hospital | 7,228 | 28.36 |
| Missing | 569 | 2.23 |
| **Facility Ownership** | | |
| Public | 20,075 | 78.77 |
| Private | 4,840 | 18.99 |
| Missing | 569 | 2.23 |
| **WHO Clinical Stage** | | |
| I | 11,712 | 45.96 |
| II | 5,594 | 21.95 |
| III | 6,409 | 25.15 |
| IV | 1,176 | 4.61 |
| Missing data | 593 | 2.33 |
| **Prior Exposure on ART** | | |
| Yes | 9,968 | 39.11 |
| No | 15,516 | 60.89 |
| **HIV/TB Co-Infection** | | |
| Yes | 366 | 1.44 |
| No | 23,296 | 91.41 |
| Missing | 1,822 | 7.15 |
| **Geographical zone** | | |
| Coastal | 5,620 | 22.05 |
| Central | 2,249 | 8.83 |
| Lake | 7,506 | 29.45 |
| Northern | 2,031 | 7.97 |
| Southern Highland | 6,632 | 26.02 |
| Western | 877 | 3.44 |
| Missing | 569 | 2.23 |
| **ART Regimens** | | |
| Firstline Regimen | 25,188 | 98.84 |
| Second line Regimen | 296 | 1.16 |
| **Nutritional Status** | | |
| Malnutrition | 94 | 0.37 |
| No Malnutrition | 23,114 | 90.70 |
| Missing | 2,276 | 8.93 |

with an overall rate of 1.33 per 1000 (95% CI; 38–40) pmr (Table 2). Rate of LTFU among adolescent on ART at 3,6,12 and 24 were 0.16, 5.60, 1.91 and 0.58 per 1000 pmr respectively. The highest rates were among adolescent aged 15–19 (1.63, 95% CI; 1.59–1.66); female adolescents (1.50, 95% CI; 1.47–1.53); with WHO stage I (1.72, 95% CI; 1.68–1.77) and residing in Lake zone (1.59, 95% CI; 1.54–1.64). Likewise, a high rate of LTFU was observed among malnourished adolescents (2.04, 95% CI; 1.52–2.74) and those in first-line regimen (1.35, 95% CI; 1.32–1.38) (Table 2). Also, the median time of LTFU from ART care was 5.59 months. Likewise, the overall failure probabilities of LTFU among adolescents on ART were 11.2%, 17% and 24% at 3, 6 and 12 months respectively. Failure probability was high among adolescents aged 15–19 years and those with HIV/TB co-infection (Fig 2).

## Predictors associated with LTFU among adolescents

In bivariate analysis predictors associated with increased risk of LTFU were; age 15–19 years (crude hazard ratio (cHR): 2.24; 95% CI, 2.13–2.35), attending clinics at dispensaries (cHR: 1.39; 95% CI, 1.33–1.46) and health centers (cHR: 1.22; 95% CI, 1.16–1.28) and attending clinics at public facilities (cHR: 2.24; 95% CI, 2.13–2.35). Also, residing in central (cHR: 1.22; 95% CI, 1.11–1.34), lake zone (cHR: 1.39; 95% CI, 1.29–1.51) and HIV/TB co-Infection (cHR: 1.29; 95% CI, 1.10–1.51).

In multivariate Cox proportion hazard analysis, predictors associated with increased risk of LTFU were age 15–19 years (aHR: 1.57; 95% CI, 1.47–1.69), HIV/TB co-infection (aHR: 1.58; 95% CI, 1.32–1.89), attending clinic at dispensary (aHR: 1.12; 95% CI, 1.07–1.18) and health centers (aHR: 1.10; 95% CI, 1.04–1.15). Likewise, attending clinics at public facilities (aHR: 1.08; 95% CI, 1.02–1.14), residing in Lake zone (aHR: 1.09; 95% CI, 1.02–1.18) and being malnourished (aHR: 2.27; 95% CI, 1.56–3.23) were associated with increased risk of LTFU. Also, increased risk of LTFU was observed among female adolescents (aHR: 1.16; 95% CI, (1.09–1.25), having WHO stage III (aHR: 1.22; 95% CI, 1.11–1.37) and stage IV (aHR: 1.22; 95% CI, 1.11–1.35). Lower risk of LTFU was noticed among patients taking second-line regimen with 40% reduced risk of LTFU compared to those on first-line regimen (aHR: 0.40; 95% CI, (0.30–0.53)) (Table 3).

## Discussion

This study bid to determine the rate and predictors of LTFU among adolescents on ART in Tanzania. We identified a high rate of LTFU among ADLHIV on ART in our study. Moreover, predictors of LTFU on ART were; HIV/TB Co-Infection, attending clinics at dispensaries, health centers, being malnourished, WHO stage II, III, and IV, Second-line ART regimen and Lake Zone.

Our study observed a high rate of loss to follow up among adolescents compared to WHO 90-90-90 targets., Nevertheless, this rate of LTFU among adolescents was observed to be low compared to the study conducted in Myanmar (69%) [19]. However compared to our results, a low proportion of loss to follow up on ART was reported in a study conducted in Ethiopia (13.6%) [20]. Also, the rate of LTFU was high during the first months of ART initiation as compared to the study conducted in Kenya, Ethiopia, and Zimbabwe [20–22]. The disparity of median time in the previous studies might be due to different times of follow up, a study conducted in Ethiopia and Kenya had a follow-up time of eight and four years respectively [20, 21]. Viral load suppression, being a mobile group (shifting of schools), fear of discrimination, and fear of disclosure is anticipated to cause dropout in the first six months of ART care. Strengthening the follow-up system and establishment of youth HIV support groups are highly encouraged.

**Table 2. Rate of loss to follow up among adolescents on ART.**

| Characteristics | Person-Month (pm) | No of loss to follow up among Adolescents on ART | Rate of Loss to follow up Per 1000pm | 95% CI | |
|---|---|---|---|---|---|
| | | | | Lower | Upper |
| **Loss to follow up rate /Tanzania Mainland** | **8,100** | **10,735** | **1.33** | **1.31** | **1.36** |
| 3 Month | 115.65 | 19 | 0.16 | 0.10 | 0.26 |
| 6 Months | 966.70 | 5418 | 5.60 | 5.46 | 5.76 |
| 12 Months | 1700 | 3194 | 1.91 | 1.84 | 1.98 |
| 24 Months | 3000 | 1744 | 0.58 | 0.56 | 0.61 |
| **Sex** | | | | | |
| Male | 1900 | 1558 | 0.80 | 0.76 | 0.84 |
| Female | 6100 | 9177 | 1.50 | 1.47 | 1.53 |
| **Age Group (Years)** | | | | | |
| 10–14 | 2600 | 1860 | 0.72 | 0.69 | 0.75 |
| 15–19 | 5500 | 8875 | 1.63 | 1.59 | 1.66 |
| **Types of Health Facility** | | | | | |
| Dispensary | 2700 | 4165 | 1.56 | 1.51 | 1.61 |
| Health Center | 2700 | 3696 | 1.36 | 1.32 | 1.41 |
| Hospital | 2500 | 2763 | 1.12 | 1.07 | 1.16 |
| Missing | 191.5 | 111 | 0.58 | 0.48 | 0.69 |
| **Facility Ownership** | | | | | |
| Public | 6200 | 8725 | 1.39 | 1.37 | 1.42 |
| Private | 1600 | 1899 | 1.17 | 1.12 | 1.23 |
| Missing | 191.50 | 111 | 0.58 | 0.48 | 0.69 |
| **Marital Status** | | | | | |
| Never Married/Single | 515 | 1,214 | 2.36 | 2.22 | 2.49 |
| Cohabiting/Married | 841 | 1,395 | 1.66 | 1.57 | 1.75 |
| Missing | 317 | 585 | 1.85 | 1.70 | 2.00 |
| **WHO Clinical Stage** | | | | | |
| I | 3400 | 5858 | 1.72 | 1.68 | 1.77 |
| II | 1800 | 2026 | 1.10 | 1.05 | 1.15 |
| III | 2300 | 2177 | 0.96 | 0.92 | 1.00 |
| IV | 406 | 453 | 1.12 | 1.02 | 1.22 |
| Missing | 138.41 | 221 | 1.59 | 1.39 | 1.82 |
| **Nutrition Status** | | | | | |
| Malnutrition | 21.59 | 44 | 2.04 | 1.52 | 2.74 |
| No Malnutrition | 7400 | 9915 | 1.35 | 1.32 | 1.37 |
| Missing | 675.82 | 776 | 1.15 | 1.07 | 1.23 |
| **ART Regime** | | | | | |
| First line Regime | 7900 | 10684 | 1.35 | 1.32 | 1.38 |
| Second line Regime | 137.52 | 51 | 0.37 | 0.28 | 0.49 |
| **HIV/TB Co-Infection** | | | | | |
| Yes | 91.77 | 154 | 1.36 | 1.33 | 1.38 |
| No | 7400 | 10084 | 1.68 | 1.43 | 1.96 |
| Missing | 539.35 | 497 | 0.92 | 0.84 | 1.00 |
| **Prior Exposure on ART** | | | | | |
| Yes | 3200 | 4021 | 1.27 | 1.23 | 1.31 |
| No | 4900 | 6714 | 1.37 | 1.34 | 1.41 |
| **Geographical Zones** | | | | | |

*(Continued)*

**Table 2.** (Continued)

| Characteristics | Person-Month (pm) | No of loss to follow up among Adolescents on ART | Rate of Loss to follow up Per 1000pm | 95% CI | |
|---|---|---|---|---|---|
| | | | | Lower | Upper |
| Central Zone | 701.47 | 975 | 1.39 | 1.31 | 1.48 |
| Coastal Zone | 1800 | 2342 | 1.29 | 1.25 | 1.35 |
| Lake Zone | 2200 | 3517 | 1.59 | 1.54 | 1.64 |
| Northern Zone | 719.79 | 819 | 1.13 | 1.06 | 1.21 |
| Southern Highland | 2200 | 2614 | 1.20 | 1.16 | 1.25 |
| Western Zone | 269.07 | 357 | 1.33 | 1.19 | 1.47 |
| Missing | 191.45 | 111 | 0.58 | 0.48 | 0.69 |

Increased risk of LTFU among adolescents with WHO stage III and IV observed in our findings was similar to the previous studies conducted in West Africa, Tanzania, Uganda, and Zambia [23, 24]. Also, studies conducted in Ethiopia and Uganda showed an increased risk of LTFU among adolescents with WHO stage III and IV [25, 26]. No association between LTFU and the WHO stage was observed in the study conducted in Kenya [21]. In our study, an increased risk of LTFU among adolescents who were in WHO stage III and IV might be due to the death of the disease unbeknownst to the health system. Patients with advanced WHO stage might have severe malnutrition or undiagnosed infections such as tuberculosis resulting in an increased mortality rate [27]. Nevertheless, patients with a worse prognosis at baseline are more likely to be loss to follow up [27]. These findings call for early identification of HIV-infected individuals and early initiation of ART.

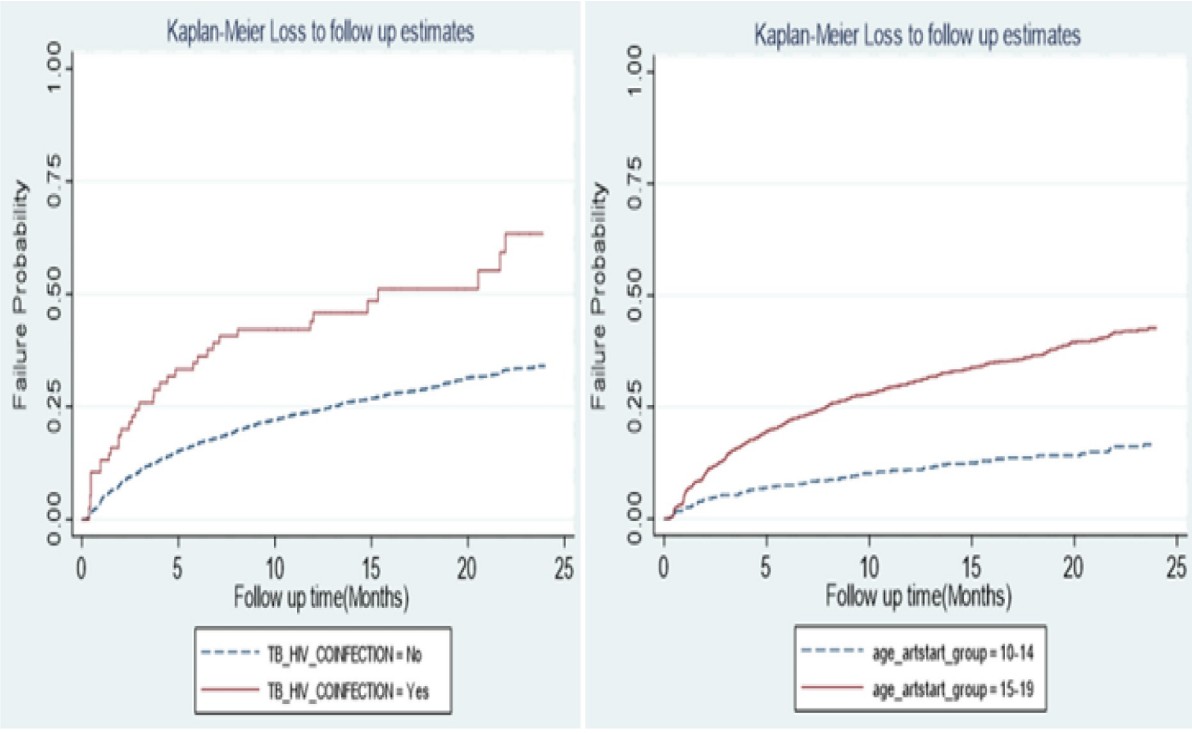

**Fig 2. Failure probability among adolescents in HIV/TB Co-infection status and age group.**

**Table 3. Bivariate and multivariate analysis of loss to follow up among adolescent on ART.**

| Variable | Bivariate analysis | | Multivariate analysis | |
| --- | --- | --- | --- | --- |
| | Crude Hazard Ratio (95%CI) | p-value | Adjusted Hazard Ratio (95% CI) | p-value |
| **Sex** | | | | |
| Male | Reference | | Reference | |
| Female | 1.89(1.79–1.96) | <0.001 | **1.16(1.09–1.25)** | **<0.001** |
| **Age Group (Years)** | | | | |
| 10–14 | Reference | | Reference | |
| 15–19 | 2.24(2.13–2.35) | <0.001 | **1.57(1.47–1.69)** | **<0.001** |
| **Types of Health Facility** | | | | |
| Hospital | Reference | | Reference | |
| Dispensary | 1.39(1.33–1.46) | <0.001 | **1.12(1.07–1.18)** | **<0.001** |
| Health Center | 1.22(1.16–1.28) | <0.001 | **1.10(1.04–1.15)** | **<0.001** |
| **Facility Ownership** | | | | |
| Private | Reference | | Reference | |
| Public | 1.18(1.13–1.25) | <0.001 | **1.08(1.02–1.14)** | **0.004** |
| **Marital Status** | | | | |
| Never married/Single | Reference | | Reference | |
| Cohabiting/Married | 0.57(0.55–0.60) | <0.001 | 0.89(0.69–1.14) | 0.347 |
| **WHO Clinical Stage** | | | | |
| I | Reference | | Reference | |
| II | 0.67(0.61–0.74) | <0.001 | 1.03(0.93–1.14) | 0.642 |
| III | 1.04(0.94–1.15) | 0.327 | **1.22(1.11–1.37)** | **<0.001** |
| IV | 1.18(1.06–1.32) | 0.013 | **1.22(1.11–1.35)** | **<0.001** |
| **Nutrition Status** | | | | |
| No Malnutrition | Reference | | Reference | |
| Malnutrition | 1.56(1.16–2.08) | **<0.001** | **2.27(1.56–3.23)** | **<0.001** |
| **ART Regimen** | | | | |
| Firstline Regimen | Reference | | Reference | |
| Secondline Regimen | 0.28(0.22–0.37) | <0.001 | **0.40(0.30–0.53)** | **<0.001** |
| **Geographical Zone** | | | | |
| Northern zone | Reference | | Reference | |
| Central Zone | 1.22(1.11–1.34) | <0.001 | 1.07(0.97–1.18) | 0.149 |
| Coastal Zone | 1.14(1.06–1.24) | 0.001 | 1.04(0.96–1.13) | 0.343 |
| Lake Zone | 1.39(1.29–1.51) | <0.001 | **1.09(1.02–1.18)** | **0.030** |
| Southern Highland Zone | 1.06(0.98–1.15) | 0.134 | **0.90(0.83–1.0.97)** | **0.010** |
| Western Zone | 1.17(1.04–1.33) | 0.012 | 0.89(0.79–1.02) | 0.100 |
| **HIV/TB Co-Infection** | | | | |
| No | Reference | | Reference | |
| Yes | 1.29(1.10–1.51) | 0.002 | **1.58(1.32–1.89)** | **<0.001** |
| **Prior Exposure on ART** | | | | |
| No | Reference | | Reference | |
| Yes | 0.92(0.88–0.96) | <0.001 | **0.90(0.87–0.94)** | **<0.001** |

Furthermore, a prominent risk of LTFU in ART care among adolescents with TB infection observed in our study corroborated with the study conducted in Ethiopia [25]. A study in sub-Saharan Africa revealed no association between LTFU and HIV/TB co-infection [28]. Increased LTFU among HIV/TB adolescents in our study might be due to medication-related issues such as adverse effects, pill burden, or complexity of drug regimen. To ensure close

follow-up among adolescents in ART care it is essential to strengthen TB screening for early diagnosis and the use of home-based care workers.

Adolescents attending clinics at dispensaries and health centers were more likely to be LTFU. Similar findings were reported in studies conducted in Uganda, rural Zimbabwe, and South Africa [14, 22, 29]. Other preceding studies elaborated adolescents who attended clinics in hospitals were more likely to dropout [25, 30]. However, a study conducted in sub-Saharan countries explained no association between type of health facility and LTFU [28]. In our study, the high risk of LTFU among adolescents at primary health facilities might be attributed to inadequate human resources resulting in long waiting time. Also, non-adherence to ART care might be caused by poor quality of adolescents' HIV services at primary health facilities. This calls for greater investment in healthcare workforce and establishment of integrated adolescents' ART clinics at the primary health facilities.

We also identified high risk of LTFU among adolescents who attended clinics at public health facilities. These findings were similar to a study conducted in Myanmar which showed an increased risk of LTFU among patients treated in public sectors [31]. However, a study conducted in Uganda explained no significant association between facility ownership and LTFU [32]. The high risk of LTFU in our results might be due to patients' workload and long waiting time at the public facilities compared to private facilities. There is a need for establishing youth-friendly ART clinics, scheduling clinic visits on school holidays, and facilitating transport to clinics. Also, follow-up through telephone calls and shortening waiting time might improve retention on ART.

Adolescents on second-line ART regimen had reduced risk of LTFU, our findings concur with a study conducted in Ethiopia [33]. However, our results contrast with the studies conducted in Myanmar [34] and Nigeria [35] which reported an increased risk of LTFU among patients on a second-line regimen. A study conducted in Nigeria stated an increased risk of LTFU might be caused by adverse effects obtained from second-line drugs [35]. The reduced risk among adolescents in the second-line found in our study could be due to the close follow-up given to this group after treatment failure.

We identified that adolescents aged 15–19 years had a higher risk of LTFU from ART care. The results corresponded to previous studies conducted in Ethiopia, South Africa, and sub-Saharan Africa [20, 30, 36]. Growing independent, fear of stigma, peer pressure, discrimination, and being a mobile group with the shifting of schools might have contributed to LTFU among older adolescents [8, 37]. Nevertheless, there is a need for conducting age-specific interventions to reduce LTFU among adolescents. The increased risk of LTFU among female adolescents observed in our study was a consistency to the previous findings in Uganda and a study conducted on MTCT-Plus programs in 9 different countries [38, 39]. However, studies in Tanzania, Ethiopia, and Malawi reported a high risk of LTFU on ART among male adolescents [40–42]. Nevertheless, a study conducted in 15 ART programs in Africa, Asia, and South America reported no association between gender and LTFU [43]. A study conducted in Kenya reported that the reasons for high LTFU among females included; family commitments, high transport costs, and work commitments [44]. In our study, the reason for gender gap is that more females accessed HIV testing services than males, especially during Antenatal Clinic (ANC) services. Therefore, the high LTFU among female adolescents in our study might be attributed to challenges of securing childcare to attend follow-up clinic visits or undocumented transfer to other HIV care clinics. This calls for strengthening the linkage to HIV care and counseling among female adolescents and sustained outreach after delivery.

It was observed that adolescents who resided in the Lake zone were more likely to be LTFU in care. The variation of LTFU risk was also reported in various studies across different geographical zones [11, 25, 45]. The differences in retention in care could be due to variations of

social, cultural, and religious practices [25, 35]. We speculated socio-cultural and economic activities such as early marriages, mining, and fishing activities might have led to a high risk of LTFU among adolescents. A study conducted in Lake zone reported that local beliefs that HIV-like illnesses were attributed to witchcraft, thus majority of HIV patients preferred traditional healers [46]. These beliefs might have resulted in poor attendance of adolescents in ART clinics. There is a need for region-specific interventions that will prioritize areas with a high risk of LTFU and ensure close follow up of adolescents in ART.

This study had several limitations due to the use of routine data from the existing database. We did not ascertain treatment failure as an outcome of LTFU from ART care using CD4 counts and viral load counts. However, a strong link between LTFU and treatment failure published in various studies in Tanzania and elsewhere provide justification for addressing LTFU among adolescents [25, 47]. Also, we were not able to assess if adolescents presented for care at other clinics during the two-year follow-up time. However, the strength of this study was the inclusion of adolescents in HIV care from the whole country making our study results generalizable to all HIV-infected adolescents on ART care in Tanzania. Additionally, the longitudinal nature of this analysis provided an opportunity to assess rate and time of LTFU and associated predictors.

## Conclusion

Due to high rate of LTFU obtained in our findings than in other studies, retention in care among adolescents in ART is still of major concern in Tanzania. There is a need for targeted interventions for adolescents; aged 15–19 years, HIV/TB co-infected, with WHO stage IV, and those residing in the Lake zone. Also, attention among adolescents attending clinics at public facilities, dispensaries, and health centers is highly needed. There is a need for designing integrated clinics for adolescents, especially in primary health facilities to increase retention in care. Generally, interventions that will develop strategies for reducing LTFU among adolescents especially in the first six months in ART are highly warranted to be able to achieve the 2030 goal of ending the HIV epidemic as a public health threat.

## Acknowledgments

We thank Tanzania National AIDS Control Program (NACP) for letting us access patients' data for this study as per the Program data access regulations. We acknowledge the valuable support received from Dr Ahmed Abade and Dr Werner in supporting the design of this study.

## Author Contributions

**Conceptualization:** Esther-Dorice Tesha, Elia Mmbaga.

**Data curation:** Esther-Dorice Tesha, Prosper Njau.

**Formal analysis:** Esther-Dorice Tesha, Baraka Revocutus.

**Funding acquisition:** Rogath Kishimba.

**Methodology:** Esther-Dorice Tesha, Elia Mmbaga.

**Writing – original draft:** Esther-Dorice Tesha, Elia Mmbaga.

**Writing – review & editing:** Esther-Dorice Tesha, Rogath Kishimba, Prosper Njau, Elia Mmbaga.

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
