## [Decision Letter · Decision Letter 0]

23 Dec 2021

PONE-D-21-31368Predictors of loss to follow up from antiretroviral therapy among adolescents with HIV/AIDS in TanzaniaPLOS ONE

Dear Dr. Tesha,

Thank you for submitting your manuscript to PLOS ONE. After careful consideration, we feel that it has merit but does not fully meet PLOS ONE’s publication criteria as it currently stands. Therefore, we invite you to submit a revised version of the manuscript that addresses the points raised during the review process.

ACADEMIC EDITOR: This is an interesting manuscript addressing an important topic of LTFU among adolescents living with HIV. However, there are several critical weaknesses identified by the reviewers that would need to be addressed before this manuscript could be considered for publication in PLoS One. The methodology needs clarification as indicated by reviewer #3. In addition, it is unclear how deaths were separated from LTFU - especially given some of the finding. Can deaths be miss-classified as LTFU? Also please clearly  define all outcomes and variables. If resubmitting please address these and the issues by the reviewers in a detailed response letter with the revised manuscript. 

We look forward to receiving your revised manuscript.

Kind regards,

Brian C. Zanoni, MD

Academic Editor

PLOS ONE

Journal Requirements:

Our special appreciation goes to Tanzania field epidemiology and Laboratory training program for its funding support.

We acknowledge the financial support from the Tanzania Field Epidemiology and Laboratory Training Program(TFELTP) during development and preparation of this study. TFELTP had no role in study design, data collection and analysis, decision to publish, or preparation of the manuscript.

Additional Editor Comments:

This is an interesting manuscript addressing an important topic of LTFU among adolescents living with HIV. However, there are several critical weaknesses identified by the reviewers that would need to be addressed before this manuscript could be considered for publication in PLoS One. The methodology needs clarification as indicated by reviewer #3. In addition, it is unclear how deaths were separated from LTFU - especially given some of the finding. Can deaths be miss-classified as LTFU? Also the definitions of variables and outcomes needs to be further clarified. If resubmitting please address these and the issues by the reviewers in a detailed response letter with the revised manuscript.

Reviewers' comments:

Reviewer's Responses to Questions

**Comments to the Author**

1. Is the manuscript technically sound, and do the data support the conclusions?

Reviewer #1: Yes

Reviewer #2: Partly

Reviewer #3: Yes

2. Has the statistical analysis been performed appropriately and rigorously? 

Reviewer #1: Yes

Reviewer #2: I Don't Know

Reviewer #3: Yes

3. Have the authors made all data underlying the findings in their manuscript fully available?

Reviewer #1: Yes

Reviewer #2: Yes

Reviewer #3: Yes

4. Is the manuscript presented in an intelligible fashion and written in standard English?

Reviewer #1: No

Reviewer #2: No

Reviewer #3: Yes

5. Review Comments to the Author

Reviewer #1: Will need some grammatical edits

Strengths: large data set; through and sound analysis,

Areas for improvement:

Major:

Explain in methods where the various settings adolescent can access care (dispensary, health center, hospital) and perhaps a line about what that means to someone not familiar with infrastructure of care delivery in Tanzania)

Discussion includes reviews of other references which support or contradict the findings. But what I want to see is thoughtful ideas or references about why each of the factors such as living in the lake zone or getting care at a dispensary vs a hospital is causing diverse outcomes with loss to follow up. In short I want more in the discussion.

Reviewer #2: Thank you for the opportunity to review this paper on loss to follow up in adolescents living with HIV.

I think it is an important issue that the authors address but I think the paper would benefit from more information being included especially in the methods section.

Methods

1. What is the HIV prevalence in the adolescent age group in Tanzania?

2. How was malnutrition defined in this population?

3. What are first and second line regimens in Tanzania?

4. Please describe in more detail the different levels of care "dispensaries" vs "clinics"

5. Please also describe how many times adolescents are seen when starting treatment.

If follow up was over two years - surely adolescents were not TB/HIV connected for two years? Was only the baseline variable taken into account or were the variables assessed at every visit? (for malnutrition too?)

6. Was viral load available.. if yes why is it not presented?

7. Were the authors able to assess if adolescents presented for care at other clinics during the two year time period?

Results

1. How many died?

2. How many transferred out?

Reviewer #3: This is an interesting article covering an important topic. Given the high rate of LTFU seen among adolescents with HIV in this study, this is clearly an important focus to help understand associations with LTFU. A major strength is the use if a national dataset from a country with a large number of adolescents living with HIV. I thought more could have been done to discuss the findings in depth and to consider directions for addressing this challenge.

I have a few comments to strengthen the manuscript further:

1. In the abstract, it would be helpful to specify and delineate bivariate and multivariate associations with LTFU. Consider removing the description of less significant associations (lines 31-32), and removing some redundant descriptions (lines 36-38). Instead, could add more to the implications of the findings in the abstract. For example, that LTFU was high, and that in multivariate analysis, LTFU was most associated with XXXX factors. These findings point to: needed resources? Funding? Interventions? Integration of care? Adolescent-friendly services? See comments below on strengthening the discussion. These can inform how the abstract could be revised to emphasize key conclusions or interpretations. It is stated that ‘novel’ interventions are needed, but perhaps strategies might include those that are not necessarily so novel. See thoughts below on the discussion. Findings may prompt consideration of strategies including: greater investments in healthcare workforce, provision of quality adolescent-friendly services, and targeting support for those with most advanced illness or TB/HIV.

2. In the introduction, the list of predictors is confusing because some include e.g. ‘sex’ or ‘age’ and ‘WHO Stage,’ where it’s not actually clear what associations have been seen previously (e.g. older adolescence? Female sex? Advanced WHO stage?). Recommend revising to be more specific about which associations have been noted previously.

3. My most important point to revise for clarity relates to the inclusion criteria. It is stated that the study included adolescents “10-19 years initiated and enrolled on ART from 2014 to 2016”. Would add clarity to understand: does this mean this cohort only included those initiating ART at age 10 onwards? Or would it include adolescents who initiated ART before age 10? It’s an important distinction because if the study focuses on those who start ART age 10 or later, the sample may be primarily made up of adolescents with horizontal infection, or those with advanced perinatal infection, and this would exclude likely most adolescents with perinatal HIV who would start ART before age 10. This would also have implications for interpreting findings and contrasting with other studies in the discussion section. I may be mistaken in how I am reading this, but having more clarity here would avoid others potentially misinterpreting the criteria as well.

4. Would add specifics regarding “TB history” and “TB/HIV co-infection” definitions. Is this referring to having a diagnosis of TB disease? Does this include others requiring TB preventive therapy? Would add the details of how this was defined, and how it would have been documented in the routine data. Further, would similarly provide a definition for how nutritional status was determined.

5. For the variable “prior exposure on ART”, is this at the time of enrollment in care? This comes back to my earlier question, if these are patients who initiated care in 2014-2016 at age 10-19.

6. For the section on bivariate analyses (starting line 136) would give the crude hazard risks, not just the p-values.

7. It’s unclear to me why WHO Stage IV was used as a reference, rather than WHO Stage I, or similarly why malnutrition was used as a reference, rather than no malnutrition. It would be preferable to use WHO Stage I as the reference, and no malnutrition as the reference. This makes interpreting the findings clearer, particularly comparing WHO stages to Stage I as baseline, rather than comparing stage II or III to a reference of Stage IV. It is also statistically preferable to use the more common state as the reference.

8. In the discussion, would comment on how to interpret the increased LTFU among WHO Stage IV. Might there be unascertained mortality in this group? Overall, I think there needs to be more discussion about LTFU among those with advanced disease and/or TB/HIV. Also, are there data that may clarify if many of those LTFU had actually passed on? And what are the implications for care programs if severe illness is a driver of LTFU (possibly via unascertained mortality)? How could this group be better addressed?

9. Recommend commenting on the increased LTFU among those with TB/HIV. Does this reflect unascertained mortality? Burdens of combined TB/HIV clinic visits and medications? A need to integrate TB/HIV services? Also, I don’t understand the sentence which states, ‘the development of other opportunistic infection among HIV/TB patients might have accelerated LTFU due to death.’ Does this refer to deaths from a different cause and not TB? Would remove this or clarify if something is being misstated here.

10. Can the authors comment on the higher LTFU among female adolescents? What factors might make them more vulnerable? Addressing the earlier point about whether this is predominantly a cohort with horizontal HIV infection would particularly point to the acute vulnerabilities that adolescent females with new HIV diagnoses may be experiencing that present barriers to care engagement.

11. Given the higher LTFU at public facilities, and at dispensaries/health centers, would elaborate more on implications for: is there a need for expanded provision of adolescent-friendly services? Is the public sector under-resourced to provide quality adolescent-friendly services? If there are high workloads on providers in the public sector (recommend revising the phrase ‘patients’ workload’ line 195), might this point toward a need for greater investments in the health sector and in the health workforce?

12. As a minor point, there are multiple places where LTFU is misspelled as LFTU. There are other minor spelling or grammatical errors that would recommend revising. For example, recommend removing the words “contrariwise” (line 163) and “unlikely” (line 188), and revising for clarity.

6. PLOS authors have the option to publish the peer review history of their article (what does this mean?). If published, this will include your full peer review and any attached files.

Reviewer #1: **Yes: **Neerav Desai

Reviewer #2: No

Reviewer #3: No

---

## [Author Response · Author response to Decision Letter 0]

11 Mar 2022

Response to editor’s and reviewers' comments 

We are grateful for the reviews provided by the editors and external reviewers of this manuscript. The comments and suggestions are thoughtful and have helped much to improve our manuscript. We have taken them into account in the revision. Please see below in blue our detailed response to comments. 

Responses to editor’s comments

Editor’s comment No 1: - Explain in methods where the various settings adolescent can access care (dispensary, health center, hospital) and perhaps a line about what that means to someone not familiar with infrastructure of care delivery in Tanzania)

Authors' response: We are thankful for the comment. We have provided information on the various settings that can access care and information on the infrastructure of care delivery in Tanzania and revised version of the Methods section reads;

Methods, Line 74-77; "In Tanzania, adolescents can access ART care in health facilities which are categorized into three levels. These levels include; primary health facilities also known as dispensaries, secondary health facilities (health centers) and tertiary health facilities (hospitals).”

Editor’s comment No 2: - Discussion includes reviews of other references which support or contradict the findings. But what I want to see is thoughtful ideas or references about why each of the factors such as living in the lake zone or getting care at a dispensary vs a hospital is causing diverse outcomes with loss to follow up. In short I want more in the discussion. 

Authors' response: 

Authors' response: We appreciate the comment. The authors have provided more information on the factors: living in the lake zone and getting care at a dispensary versus hospitals and the revised version reads: 

Discussion, Line 288-290; “A study conducted in Lake zone reported that local beliefs that HIV-like illnesses were attributed to witchcraft, thus the majority of HIV patients preferred traditional healers[47]. These beliefs might have resulted in poor attendance of adolescents in ART clinics”.

Discussion, Line 240-245; “In our study, the high risk of LTFU among adolescents at primary health facilities might be attributed to inadequate human resources resulting in long waiting times. Also, non-adherence to ART care might be caused by the poor quality of adolescents’ HIV services at primary health facilities. This calls for greater investment in the healthcare workforce and the establishment of integrated adolescents’ ART clinics at primary health facilities.”

Editor’s comment No 3: - The methodology needs clarification as indicated by reviewer #3. In addition, it is unclear how deaths were separated from LTFU - especially given some of the findings. Can deaths be miss-classified as LTFU?-

Authors' response: Thank you for the comment. We have provided information on how deaths were separated from LTFU and newly added information reads:

Methods, Line 67-68; “Adolescents who were transferred out and died were censored at the time of their last visit.”

Editor’s comment No 4: Also please clearly define all outcomes and variables.

Authors' response: We are grateful for the comment. We have defined all outcomes and variables and the updated paragraphs reads: 

Methods, Line 95-120; " The loss to follow up was defined as any of the adolescents who failed to attend at least one clinic visit within 90 days of their scheduled appointment. Retention to care was defined as the state of adolescents being alive, actively attending scheduled appointments at the clinics and receiving ART care at the end of follow up period. Independent variables of the study included; sex which was classified as male and female, age was grouped as 10-14 years and 15-19 years. Marital status was grouped as never married/single and cohabiting/married, types of health facilities were categorized as dispensaries, health centers, hospitals. Also, facility ownership was classified as public and private, WHO clinical stages were grouped as stage I, II, III, and IV. Geographical zones were grouped as coastal, central, lake, northern, southern highland and western zones. 

Likewise, nutrition status was categorized into two groups according to BMI scales. BMI was obtained from adolescents’ weight in kilograms divided by the square of height in meters which were assessed at every clinic visit. The BMI scale of <18.5kg/m2 (underweight) and 25.0-29.9kg/m2 (obesity/overweight) were classified as malnutrition, while BMI scale of 18.5-24.9kg/m2 (normal weight) was classified as no malnutrition. ART regimens were categorized as first-line and second-line based on the Tanzania HIV guideline[19]. First-line regimens included; Tenofovir (TDF) 300 mg / Lamivudine (3TC) 300 mg / Efavirenz (EFV) 600mg. Alternative first line regimen used were; Tenofovir (TDF) + Emtricitabine (FTC) + Dolutegravir (DTG), Abacavir (ABC) + Lamivudine (3TC) + Efavirenz (EFV) and Zidovudine (AZT)+Lamivudine(3TC) +Nevirapine (NVP). Also the second-line regimens included; Zidovudine (AZT), Tenofovir (TDF), Abacavir (ABC), Lamivudine (3TC), Emtricitabine (FTC), Atazanavir boosted by Ritonavir (ATV/r), Lopinavir boosted by Ritonavir (LPV/r) and Dolutegravir (DTG). Other independent variables included; prior exposure to ART during enrolment (yes or no) and HIV/TB Co-infection (yes or no).”

Editor’s comment No 5: You indicated that you had ethical approval for your study. In your methods section, please ensure you have also stated whether you obtained consent from parents or guardians of the minors included in the study or whether the research ethics committee or IRB specifically waived the need for their consent.-

Authors' response: Thank you for the comment. We have revised the ethical section under methods and updated the paragraph reads:

Methods, Line 136-138;”This retrospective record analysis utilized anonymized data hence the institutional review board (IRB) waived the need for consent from parents and guardians of the adolescents.”

Editor’s comment No 6: Thank you for stating the following in the Acknowledgments Section of your manuscript: Our special appreciation goes to Tanzania field epidemiology and Laboratory training program for its funding support. Please note that funding information should not appear in the Acknowledgments section or other areas of your manuscript. We will only publish funding information present in the Funding Statement section of the online submission form. 

Please remove any funding-related text from the manuscript and let us know how you would like to update your Funding Statement. Currently, your Funding Statement reads as follows: We acknowledge the financial support from the Tanzania Field Epidemiology and Laboratory Training Program (TFELTP) during development and preparation’ of this study. TFELTP had no role in study design, data collection and analysis, decision to publish, or preparation of the manuscript.-

Authors' response: We appreciate the comment. We have revised acknowledgment section and the newly updated paragraph reads: 

Acknowledgment, Line 317-319; “We thank the Tanzania National AIDS Control Program (NACP) for letting us access patients’ data for this study as per the Program data access regulations. We acknowledge the valuable support received from Dr Ahmed Abade and Dr Werner in supporting the design of this study.”

Additional Editor’s comment No 7: This is an interesting manuscript addressing an important topic of LTFU among adolescents living with HIV. However, there are several critical weaknesses identified by the reviewers that would need to be addressed before this manuscript could be considered for publication in PLoS One. The methodology needs clarification as indicated by reviewer #3. In addition, it is unclear how deaths were separated from LTFU - especially given some of the findings. Can deaths be miss-classified as LTFU? Also, the definitions of variables and outcomes need to be further clarified.

Authors' response: Thank you for the comment. We have provided information on how deaths were separated from LTFU and define all variables and outcomes this has been answered in comment 3 and 4 above.

Responses to reviewer number 1

Reviewer comment: Explain in methods where the various settings adolescent can access care (dispensary, health center, hospital) and perhaps a line about what that means to someone not familiar with infrastructure of care delivery in Tanzania)

Authors' response: We appreciate the comment. We have provided information on the various settings that can access care and information on the infrastructure of care delivery in Tanzania and revised version reads; 

Methods, Line 74-77; "In Tanzania, adolescents can access ART care in health facilities which are categorized into three levels. These levels include; primary health facilities also known as dispensaries, secondary health facilities (health centers) and tertiary health facilities (hospitals).”

Reviewer comment: Discussion includes reviews of other references which support or contradict the findings. But what I want to see is thoughtful ideas or references about why each of the factors such as living in the lake zone or getting care at a dispensary vs a hospital is causing diverse outcomes with loss to follow up. In short, I want more in the discussion.

Authors' response: We appreciate the comment. The authors have provided more information on the factors: living in the lake zone and getting care at a dispensary versus hospitals and the revised version reads: 

Discussion, Line 288-290; “A study conducted in Lake zone reported that local beliefs that HIV-like illnesses were attributed to witchcraft, thus the majority of HIV patients preferred traditional healers[47]. These beliefs might have resulted in poor attendance of adolescents in ART clinics”.

Discussion, Line 240-245; “In our study, the high risk of LTFU among adolescents at primary health facilities might be attributed to inadequate human resources resulting in long waiting times. Also, non-adherence to ART care might be caused by the poor quality of adolescents’ HIV services at primary health facilities. This calls for greater investment in the healthcare workforce and the establishment of integrated adolescents’ ART clinics at primary health facilities.”

Responses to reviewer number 2

Reviewer comment: - What is the HIV prevalence in the adolescent age group in Tanzania?

Authors' response: We are thankful for the comment. We have added the information on the HIV prevalence among adolescents in Tanzania in the Methods section and the added sentences reads:

Methods, Line 72-74; " In 2020, population of adolescents aged 10-19 years in Tanzania mainland was projected to be 13,206,921 based on the 2012 national census[17] with an HIV prevalence of 5.8%[18]. " 

Reviewer comment: -How was malnutrition defined in this population?

Authors' response: We appreciate the comment. The authors have defined malnutrition and the new version reads; 

Methods, Line 106-110; “Likewise, nutrition status was categorized into two groups according to BMI scales. BMI was obtained from adolescents’ weight in kilograms divided by the square of height in meters which were assessed at every clinic visit. Participants with BMI of <18.5kg/m2 (underweight) or 25.0-29.9kg/m2 (obesity/overweight) were classified as “malnutrition”, while those with BMI between 18.5-24.9kg/m2 (normal weight) were classified as “no malnutrition” 

Reviewer comment: - What are first and second-line regimens in Tanzania?

Authors' response: Thank you for the comment. We have revised the text and explained first and second-line regimens in Tanzania and updated paragraph reads:

Methods, Line 112-120; “ART regimens were categorized as first-line and second-line based on the Tanzania HIV guideline[19]. First-line regimens included; Tenofovir (TDF) 300 mg / Lamivudine (3TC) 300 mg / Efavirenz (EFV) 600mg. Alternative first line regimen used were; Tenofovir (TDF) + Emtricitabine (FTC) + Dolutegravir (DTG), Abacavir (ABC) + Lamivudine (3TC) + Efavirenz (EFV) and Zidovudine (AZT)+Lamivudine(3TC) +Nevirapine (NVP). Also the second-line regimens included; Zidovudine (AZT), Tenofovir (TDF), Abacavir (ABC), Lamivudine (3TC), Emtricitabine (FTC), Atazanavir boosted by Ritonavir (ATV/r), Lopinavir boosted by Ritonavir (LPV/r) and Dolutegravir (DTG). Other independent variables included; prior exposure to ART during enrolment (yes or no) and HIV/TB Co-infection (yes or no).” 

Reviewer comment: Please describe in more detail the different levels of care "dispensaries" vs "clinics

 Authors' response: We are grateful for your valuable comments. We have described in detailed the different levels of care “dispensaries vs clinics”, the revised version now reads;

Methods, Line 74-77; "In Tanzania, adolescents can access ART care in health facilities which are categorized into three levels. These levels include; primary health facilities also known as dispensaries, secondary health facilities (health centers) and tertiary health facilities (hospitals).”

Reviewer comment: Please also describe how many times adolescents are seen when starting treatment.

Authors' response: We are grateful for the valuable comments. The authors have described on how many times adolescents are seen when starting treatment. The new sentences now read: 

Methods, Line 77-79; “The Tanzania HIV guideline requires adolescents to attend HIV clinics at least once a month for the first three months of the ART treatment and thereafter once every three months depending on their adherence status[19].

Reviewer comment: If follow-up was over two years - surely adolescents were not TB/HIV connected for two years? Was only the baseline variable taken into account or were the variables assessed at every visit? (for malnutrition too?)

Authors' response: We appreciate the comment. The authors have provided information on how TB/HIV co-infection variable and malnutrition variables were assessed, the revised version reads; 

Methods, Line 82-84; “. In Tanzania, all people living with HIV (PLHIV) are screened for TB on every clinic visit to prevent them from developing active TB by providing Isonized Preventive Therapy (IPT).” 

As already described in the Methods, 106-110; nutrition status is assessed at every clinic visit.

Reviewer comment: Was viral load available if yes why is it not presented?

Authors' response: Thank you very much for your comment. We acknowledge that we did not have comprehensive data on viral load which would have been useful as we are assessing the ultimate outcome of LTFU which is treatment failure. We have described this as a limitation in the following text added into the revised version 

Discussion, Line 294-298; “We did not ascertain treatment failure as an ultimate outcome of LTFU from ART care using CD4 counts and viral load counts. However, a strong link between LTFU and treatment failure published in various studies in Tanzania and elsewhere provide justification for addressing LTFU among adolescents[26,48].” 

Reviewer comment: Were the authors able to assess if adolescents presented for care at other clinics during the two year time period?-

Authors' response: We are grateful for the comment. The authors were not able to assess if adolescents presented for care at other clinics during the follow up time of two years. Therefore, we have revised the discussion part and explain it as part of limitation of the study. 

Discussion, Line 298-299; "Also, we were not able to assess if adolescents presented for care at other clinics during the two years follow-up time.”

Reviewer comment: How many died?

Authors' response: We appreciate the comment. The authors have provided data on how many adolescents died in the results section.

Results, Line 143-145; “About 177(0.7%) records were deceased and 74(1.7%) records were transferred out on their last appointment date.”.”

Reviewer comment: How many transferred out? 

Authors' response: We appreciate the comment. The authors have provided data on how many adolescents died in the results section as described in the previous comment.

Responses to reviewer number 3

Reviewer comment: - In the abstract, it would be helpful to specify and delineate bivariate and multivariate associations with LTFU. Consider removing the description of less significant associations (lines 31-32), and removing some redundant descriptions (lines 36-38). Instead, could add more to the implications of the findings in the abstract. For example, that LTFU was high, and that in multivariate analysis, LTFU was most associated with XXXX factors. These findings point to: needed resources? Funding? Interventions? Integration of care? Adolescent-friendly services? See comments below on strengthening the discussion. These can inform how the abstract could be revised to emphasize key conclusions or interpretations. It is stated that ‘novel’ interventions are needed, but perhaps strategies might include those that are not necessarily so novel. See thoughts below on the discussion. Findings may prompt consideration of strategies including: greater investments in healthcare workforce, provision of quality adolescent-friendly services, and targeting support for those with most advanced illness or TB/HIV.

Authors' response: We are thankful for the comment. We have added information on the implications of the findings, the revised now reads:

Abstract, line 28-35 “Predictors associated with LTFU included; adolescents aged 15-19 years (adjusted hazard ratio (aHR): 1.57; 95% Confidence Interval (CI); 1.47-1.69), having HIV/TB co-infection (aHR: 1.58; 95% CI, 1.32-1.89), attending care at a dispensary (aHR: 1.12; 95% CI, 1.07-1.18) or health center (aHR: 1.10; 95% CI, 1.04-1.15), and being malnourished (aHR: 2.27; 95% CI,1.56-3.23). Moreover, residing in lake zone and having advanced HIV disease were associated with LTFU. These findings highlight the high rate of LTFU and the need for intervention targeting older adolescents with advanced disease and strengthening primary public facilities in order to achieve the 2030 goal of ending HIV as a public health threat.” 

Reviewer comment: - In the introduction, the list of predictors is confusing because some include e.g. ‘sex’ or ‘age’ and ‘WHO Stage,’ where it’s not actually clear what associations have been seen previously (e.g. older adolescence? Female sex? Advanced WHO stage?). Recommend revising to be more specific about which associations have been noted previously.-

Authors' response: Thank you for your comment. We have revised the list of predictors and shows which associations were noted previously and the newly paragraph reads:

Introduction, Line 46-51; “Predictors associated with increased risk of LTFU among ADLHIV reported in various studies included; ADLHIV aged 15-19 years, female adolescents, those diagnosed with HIV/TB co-infection, those with malnutrition, adolescents who attended clinics at primary facilities and having advanced WHO clinical stage [7–13]. Also increased risk of LTFU was observed among adolescents who had prior exposure to ART[14] and those who attended clinics at public health facilities[15].”

Reviewer comment: - My most important point to revise for clarity relates to the inclusion criteria. It is stated that the study included adolescents “10-19 years initiated and enrolled on ART from 2014 to 2016”. Would add clarity to understand: does this mean this cohort only included those initiating ART at age 10 onwards? Or would it include adolescents who initiated ART before age 10? It’s an important distinction because if the study focuses on those who start ART age 10 or later, the sample may be primarily made up of adolescents with horizontal infection or those with advanced perinatal infection, and this would exclude likely most adolescents with perinatal HIV who would start ART before age 10. This would also have implications for interpreting findings and contrasting with other studies in the discussion section. I may be mistaken in how I am reading this, but having more clarity here would avoid others potentially misinterpreting the criteria as well.-

Authors' response: Thank you for the comment. We have revised the inclusion criteria and exclusion criteria and the new paragraph reads: 

Methods, Line 63-65; “This study included adolescents who were enrolled and initiated ART at age 10-19 with either vertical or horizontal HIV/AIDS infection from January 2014 to December 2016.”

Reviewer comment: - Would add specifics regarding “TB history” and “TB/HIV co-infection” definitions. Is this referring to having a diagnosis of TB disease? Does this include others requiring TB preventive therapy? Would add the details of how this was defined, and how it would have been documented in the routine data. Further, would similarly provide a definition for how nutritional status was determined.-

Authors' response: We are grateful for the comments. We have revised the document and added the definition of TB/HIV co-infection and that of nutritional status.

Methods, Line 81-84; “TB/HIV Co-infection refers to HIV patients who were also diagnosed with TB infection during routine TB screening. In Tanzania, all people living with HIV (PLHIV) are screened for TB on every clinic visit and those not infected are provided with Isonized Preventive Therapy (IPT) to prevent them from developing active TB”.

Methods, Line 108-110; “Participants with BMI of <18.5kg/m2 (underweight) or 25.0-29.9kg/m2 (obesity/overweight) were classified as “malnutrition”, while those with BMI between 18.5-24.9kg/m2 (normal weight) were classified as “no malnutrition”.

Reviewer comment: For the section on bivariate analyses (starting line 136) would give the crude hazard risks, not just the p-values.-

Authors' response: We appreciate the comment. We have revised the bivariate analyses and have added the crude hazard risks and newly updated paragraphs reads. 

Results, Line 179-184; “In bivariate analysis predictors associated with increased risk of LTFU were; age 15-19 years (crude hazard ratio (cHR): 2.24; 95% CI, 2.13-2.35),, attending clinics at dispensaries (cHR: 1.39; 95% CI, 1.33-1.46)and health centers (cHR: 1.22; 95% CI, 1.16-1.28)and attending clinics at public facilities (cHR: 2.24; 95% CI, 2.13-2.35). Also, residing in central (cHR: 1.22; 95% CI, 1.11-1.34), lake zone (cHR: 1.39; 95% CI, 1.29-1.51) and HIV/TB co-Infection (cHR: 1.29; 95% CI, 1.10-1.51)”. 

Reviewer comment: It’s unclear to me why WHO Stage IV was used as a reference, rather than WHO Stage I, or similarly why malnutrition was used as a reference, rather than no malnutrition. It would be preferable to use WHO Stage I as the reference, and no malnutrition as the reference. This makes interpreting the findings clearer, particularly comparing WHO stages to Stage I as baseline, rather than comparing stage II or III to a reference of Stage IV. It is also statistically preferable to use the more common state as the reference.

Authors' response: We are grateful for the valuable comments. We have revised the analysis and kept WHO stage I as the reference, and no malnutrition as the reference. This has been accommodated in table 3“Bivariate and Multivariate analysis of loss to follow up among Adolescents on ART”

 

Reviewer comment: In the discussion, would comment on how to interpret the increased LTFU among WHO Stage IV. Might there be unascertained mortality in this group? Overall, I think there needs to be more discussion about LTFU among those with advanced disease and/or TB/HIV. Also, are there data that may clarify if many of those LTFU had actually passed on? And what are the implications for care programs if severe illness is a driver of LTFU (possibly via unascertained mortality)? How could this group be better addressed?

Authors' response: We are thankful for the comment. We have revised the discussion about the increased risk of LTFU among WHO stage IV, and the newly paragraph reads; 

Discussion, Line 220-227; “No association between LTFU and the WHO stage was observed in the study conducted in Kenya[22]. In our study, an increased risk of LTFU among adolescents who were in WHO stage III and IV might have been caused by unbeknownst death to the health system. Patients with advanced WHO stage might have severe malnutrition or undiagnosed infections such as tuberculosis resulting in an increased mortality rate[28]. Nevertheless, patients with a worse prognosis at baseline are more likely to be loss to follow up[28]. These findings calls for early identification of HIV-infected individuals and early initiation of ART.”

Reviewer comment: Recommend commenting on the increased LTFU among those with TB/HIV. Does this reflect unascertained mortality? Burdens of combined TB/HIV clinic visits and medications? A need to integrate TB/HIV services? Also, I don’t understand the sentence which states, ‘the development of other opportunistic infection among HIV/TB patients might have accelerated LTFU due to death.’ Does this refer to deaths from a different cause and not TB? Would remove this or clarify if something is being misstated here.

Authors' response: We are thankful for the comment. We have revised the discussion about the increased risk of LTFU among those with TB/HIV, and newly added information reads; 

Discussion, Line 230-234; “Increased LTFU among HIV/TB adolescents in our study might be due to medication-related issues such as adverse effects, pill burden or complexity of drug regimen. To ensure close follow up among adolescents in ART care it is essential to strengthen TB screening for early diagnosis and the use of home-based care workers.”

Reviewer comment: Can the authors comment on the higher LTFU among female adolescents? What factors might make them more vulnerable? Addressing the earlier point about whether this is predominantly a cohort with horizontal HIV infection would particularly point to the acute vulnerabilities that adolescent females with new HIV diagnoses may be experiencing that present barriers to care engagement.

Authors' response: We are appreciate for the comment. We have revised the discussion and we have commented on the higher LTFU among female adolescents and the new paragraph reads; 

Discussion, Line 269-281; “The increased risk of LTFU among female adolescents observed in our study was consistency with the previous findings in Uganda and a study conducted on MTCT-Plus programs in 9 different countries[39,40]. However studies in Tanzania, Ethiopia and Malawi reported a high risk of LTFU on ART among male adolescents[41–43]. Nevertheless, a study conducted in 15 ART programs in Africa, Asia, and South America reported no association between gender and LTFU [44]. A study conducted in Kenya reported that the reasons for high LTFU among females included family commitments, high transport costs and work commitments[45].In our study, the reason for the gender gap is that more females accessed HIV testing services than males, especially during Antenatal Clinic (ANC) services. Therefore, the high LTFU among female adolescents in our study might be attributed to challenges of securing childcare to attend follow-up clinic visits or undocumented transfer to other HIV care clinics. This calls for strengthening the linkage to HIV care and counselling among female adolescents and sustained outreach after delivery.”

Reviewer comment: As a minor point, there are multiple places where LTFU is misspelled as LFTU. There are other minor spelling or grammatical errors that would recommend revising. For example, recommend removing the words “contrariwise” (line 163) and “unlikely” (line 188), and revising for clarity.

Authors' response: We are appreciate for the comment. We have revised the whole manuscript on the grammatical errors.

---

## [Decision Letter · Decision Letter 1]

10 May 2022

Predictors of loss to follow up from antiretroviral therapy among adolescents with HIV/AIDS in Tanzania

PONE-D-21-31368R1

Dear Dr. Tesha,

We’re pleased to inform you that your manuscript has been judged scientifically suitable for publication and will be formally accepted for publication once it meets all outstanding technical requirements.

Kind regards,

Brian C. Zanoni, MD

Academic Editor

PLOS ONE

Additional Editor Comments (optional):

The authors have appropriately responded to the comments of the editor and of the reviewers. The manuscript is acceptable for publication. However, I suggest the authors carefully review the manuscript and correct several language, grammar and punctuation errors throughout the manuscript.

Reviewers' comments:

Reviewer's Responses to Questions

**Comments to the Author**

1. If the authors have adequately addressed your comments raised in a previous round of review and you feel that this manuscript is now acceptable for publication, you may indicate that here to bypass the “Comments to the Author” section, enter your conflict of interest statement in the “Confidential to Editor” section, and submit your "Accept" recommendation.

Reviewer #1: All comments have been addressed

Reviewer #2: All comments have been addressed

2. Is the manuscript technically sound, and do the data support the conclusions?

Reviewer #1: Yes

Reviewer #2: Yes

3. Has the statistical analysis been performed appropriately and rigorously? 

Reviewer #1: Yes

Reviewer #2: Yes

4. Have the authors made all data underlying the findings in their manuscript fully available?

Reviewer #1: Yes

Reviewer #2: Yes

5. Is the manuscript presented in an intelligible fashion and written in standard English?

Reviewer #1: Yes

Reviewer #2: Yes

6. Review Comments to the Author

Reviewer #1: They have addressed the issues I was concerned with. I noticed several grammatical errors in the added sections. Please fix several grammatical mistakes.

Reviewer #2: Thank you for addressing the comments and for clarifications and revising the manuscript

I am satisfied with the responses.

7. PLOS authors have the option to publish the peer review history of their article (what does this mean?). If published, this will include your full peer review and any attached files.

Reviewer #1: **Yes: **Neerav Desai MD

Reviewer #2: No

---

## [Editor Report · Acceptance letter]

29 Jun 2022

PONE-D-21-31368R1 

Predictors of loss to follow up from antiretroviral therapy among adolescents with HIV/AIDS in Tanzania 

Dear Dr. Tesha:

I'm pleased to inform you that your manuscript has been deemed suitable for publication in PLOS ONE. Congratulations! Your manuscript is now with our production department. 

Kind regards, 

on behalf of

Dr. Brian C. Zanoni 

Academic Editor

PLOS ONE